# A Simple Baseline for Predicting Events with Auto-Regressive Tabular Transformers

## Abstract

Many real-world applications of tabular data involve using historic events to predict properties of new ones, for example whether a credit card transaction is fraudulent or what rating a customer will assign a product on a retail platform. Existing approaches to event prediction include costly, brittle, and application-dependent techniques such as time-aware positional embeddings, learned row and field encodings, and oversampling methods for addressing class imbalance. Moreover, these approaches often assume specific use-cases, for example that we know the labels of all historic events or that we only predict a pre-specified label and not the data's features themselves. In this work, we propose a simple but flexible baseline using standard autoregressive LLM-style transformers with elementary positional embeddings and a causal language modeling objective. Our baseline outperforms existing approaches across popular datasets and can be employed for various use-cases. We demonstrate that the same model can predict labels, impute missing values, or model event sequences.

## 1 Introduction

What does the past tell us about the present? What will happen in the future? These are fundamental questions pervasive in many scientific domains. While generative models have shown remarkable performance in language, vision, audio and even graphs, the study of predicting *events* remains underexplored. Event data consists of sequences of discrete samples that arrive at sporadic and uneven intervals. These elements can be univariate, but can also contain multiple features describing the event, or collected concurrently with the event.

Event data shares properties with other modalities, but it is unique in that the ordering and timing of events may contain important information. Unlike time-series data, which is commonly represented as uniformly spaced measurements, the time between events is important, and these intervals can vary drastically within a single event sequence. Additionally, event data shares many of the same challenges as tabular data, including heterogeneous fields (columns) that often span diverse data types (Luetto et al., 2023). However, unlike traditional tabular data, the ordering and timing of samples in event data is a primary consideration in prediction tasks. Event data can be viewed as a special case of sequential data (e.g. language) or as a special case of tabular data. However, event prediction models are often used to forecast not only properties of future events, but also when they will occur. Therefore it is not sufficient to use existing tabular or time-series specific models for event prediction; event stream data should be studied separately.

Recent methods have attempted to model event data by adapting tabular learning methods, like boosted trees, to handle additional features containing temporal information (Kang & McAuley, 2018). However, recent advances in deep learning models for tabular data (Padhi et al., 2021; Luetto et al., 2023; Gorishniy et al., 2022; Somepalli et al., 2021; Zhang et al., 2023) have enabled researchers to add contextual information to tables, including temporal information.

While these approaches have demonstrated that sequential tabular data can be modeled using deep learning and that neural networks can outperform non-parametric models, the models in this sparsely explored field are often overly complex, in both architecture and training routine. Further, existing deep learning approaches are trained specifically for a particular prediction task and must be finetuned for new tasks on the same dataset, for example predicting a new event attribute.

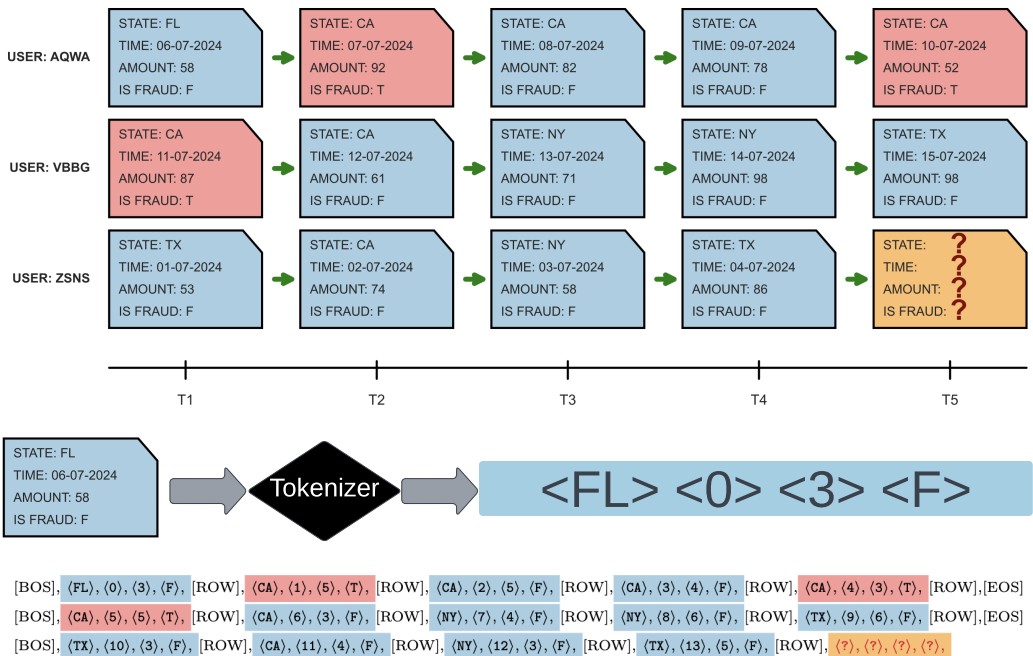

Figure 1: **Event Data Pipeline**:
**top**: STEP accepts sequences of discrete events as inputs (in this case, credit card transactions) and predicts the next token/label in a sequence. **middle**: Each event is broken into a string of tokens, with each feature represented by a distinct token. STEP models one feature at a time. Note that while the features of an event occur at the same time, this tokenization scheme adds a causal bias that must be handled by the model. **bottom**: After tokenization, event sequences are packed for training with an [EOS] token separating sequences, just as text is packed for training an autoregressive LLM.

In this work, we bring recent ideas in casual language modeling to bear for processing sequential tabular data with a simplified pipeline. Our **S**imple **T**ransformer for **E**vent **P**rediction (STEP) is a decoder-only auto-regressive model that surpasses state-of-the-art methods on popular event prediction datasets. We depart from prior methods in the following key ways:

- Our approach converts event data into a sequence of tokens to be trained with a causal language modeling loss. This makes our approach relatively simple compared to hierarchical Masked Language Modeling (MLM) pipelines specially tailored to event data. We observe that causal models outperform MLM models on a range of event prediction tasks.

- We find that causal models perform best when using standard positional encoding to represent event ordering. We depart from existing methods (Zhang et al., 2023) that use specialized positional encoding for temporal information; instead we represent time as a feature.

- We introduce two data augmentations during training that enable STEP to perform useful event prediction tasks without complicating the training pipeline and architecture: (1) We shuffle the columns within a event, allowing the model to infer unknown features of an event conditioned on any subset of observed features, a task usually unique to MLM models. (2) We mask labels from previous events, reflecting the fact that we may not know the labels of previous events while predicting the label of a new event.

## 2 RELATED WORK

Structured data is commonly found in tabular form, with each row representing a sample and each column representing a feature. This type of data is pervasive in industrial settings but only recently

has become of interest to the deep learning community. Traditional and still predominant methods for supervised and semi-supervised learning on tabular datasets involve non-parametric tree-based models like XGBoost (Chen & Guestrin, 2016), CatBoost (Prokhorenkova et al., 2019), and LightGBM (Ke et al., 2017). These models are highly competitive in performance metrics (e.g., accuracy, AUC) and offer additional benefits such as interpretability and effectiveness in both high and low data regimes.

One disadvantage of tree-based models is that they do not explicitly model inter-row dependencies, which are common in event data where rows represent sequential events, such as customer transactions over time. Attention-based "tabular transformers" have been proposed recently to address this disadvantage. These transformers differ in (1) the way that they employ the attention mechanism, (2) the type of positional encoding used, (3) the encoding of numerical features, and (4) the training regime (masked language modeling vs. causal language modeling). For example, two early works, TabNet (Arik & Pfister, 2019) and TabTransformer (Huang et al., 2020), use field-level (intra-row) attention whereas SAINT (Somepalli et al., 2021), TabBERT (Padhi et al., 2021), FATA-Trans (Zhang et al., 2023) and UniTTab (Luetto et al., 2023) employ both intra-row and inter-row attention which enables them to deal with sequential data where rows can be dependent on one another. Most of the tabular transformers use a masked language modeling approach whereas TabGPT (Padhi et al., 2021) uses a causal language modeling approach for generating synthetic data by predicting future events from an ordered sequence of 10 historical events.

Prior work has explored ways to handle specialized features present in event data, such as temporal and numeric features. In language modeling, positional encoding is used to model the order of the tokens. While most tabular models (e.g., TabBERT (Padhi et al., 2021), UniTTab (Luetto et al., 2023)) use a similar technique to capture the order of the events, FATA-Trans uses a time-aware positional embedding mechanism to also capture the non-uniform time interval between events. To address numeric features, researchers have used both a quantization approach where continuous values are discretized into bins (e.g., TabBERT) as well as learning a dense representation using a neural net (e.g., FATA-Trans, SAINT). UniTTab (Luetto et al., 2023) represents numerical values as feature vectors derived from various frequency functions. Azabou et al. (2023) model events (specifically neural brain activity) using a novel tokenization scheme combined with a pretrained model that has been finetuned to predict future neural activity.

Transformer-based approaches have been used to improve performance on time series tasks. PatchTST (Nie et al., 2023) subdivides time series into overlapping patches, treating each patch as a token for the transformer, with multivariate series handled as independent univariate series sharing the same embeddings. iTransformer (Liu et al., 2024) embeds entire time series of each variate as tokens, applying self-attention and feed-forward networks for series representation. TimesFM (Das et al., 2024) uses patches and MLP blocks with residual connections for tokenization, while Gruver et al. (2023) argue that foundation models like GPT-3 can serve as zero-shot forecasters by tokenizing data as numerical strings.

Temporal Point Processes (TPPs) have become a standard approach for modeling event sequences in continuous time. EasyTPP (Xue et al., 2024) offers an open benchmark with various datasets and neural TPP algorithms (e.g., (Shchur et al., 2020; Yang et al., 2022; Zhang et al., 2020)) for comparative evaluation. MOTOR (Steinberg et al., 2023) is a foundational time-to-event model designed for structured medical records. Additionally, EventStreamGPT (ESGPT) (McDermott et al., 2023) is an open-source toolkit specifically for modeling event stream data. Importantly, none of these methods are decoder-only, as each encodes the events before passing the learned embeddings to a model (generally a decoder) for prediction.

## 3 METHODOLOGY

### 3.1 PROBLEM DEFINITION

We consider event data, represented as tabular data with a temporal feature and arranged sequentially such that the rows are first grouped by a meta-feature (such as user) and then organized by time. Because our goal is to predict a user's future events based on their previous events, it is imperative that we separate the data by user so that events in a sequence correspond to precisely this task. We define the data as follows:

$$x_j = [f_0^{\text{meta}}, f_1^{\text{time}}, f_2, ..., f_k], \tag{1}$$

where $x_j$ is a record in the table that represents an event composed of $k + 1$ features, of which one is a meta-feature (e.g. a user ID denoted $f_0^{\text{meta}}$) and at least one is a temporal feature (i.e. a timestamp denoted $f_1^{\text{time}}$). Meta-features are columns that are not used as model inputs during training, but instead identify which rows should be grouped together (e.g. rows belonging to the same user sequence). For a given meta-feature value $u_i \in U$, all $x_j$ where $f_0^{\text{meta}} = u_i$ are arranged contiguously and in order of the temporal feature $f_1^{\text{time}}$. We denote the $n$th such training sequence of length $l$ as

$$X_n = [x_j, x_{j+1}, ..., x_{j+l-1}], \tag{2}$$

where each $x_j$ in the sequence has the same value for $f_0^{\text{meta}}$.

While it is natural for the target label to be oriented as the last feature of each event (i.e. $f_k$), this is not a strict requirement. Similarly, for a given dataset, the target label location may vary from task to task. Therefore, the model should be capable of accepting each record with the features enumerated in any order. This functionality provides two primary benefits. Firstly, the model can be evaluated with any of the columns as the target label. Specifically, any feature of $x_j$ ($[f_1^{\text{time}}, f_2, ..., f_k]$), can be designated as the target feature during evaluation. Secondly, when predicting the label for the target event, the model can be evaluated with any number of the input features (and in any order). For example, if the target label is $f_k$, any number of $[f_1^{\text{time}}, f_2, ..., f_k - 1]$ can be passed in before querying the model to predict $f_k$.

## 3.2 STEP: A SIMPLE TRANSFORMER FOR EVENT PREDICTION

We introduce **STEP**: a **S**imple **T**ransformer for **E**vent **P**rediction. Like the above methods, we use a transformer-based architecture to learn a rich semantic representation of the data, however our construction is decoder-only. Decoder-only transformers have demonstrated a remarkable ability to model sequences of tokens – we rely on that ability to model the sequential relationship between different records within a sequence of events.

In order utilize the causal masking present in these transformers, we first convert event data into the language space, by unrolling records of event data into sequences and tokenizing the resulting rows before passing the sequences into the model. As with prior methods, we quantize numeric data and arrange the data into sequences by user. Because we treat our tabular data like language, we arrange our data table in a manner similar to common language modeling setups:

1. We tokenize each feature separately by column to ensure that entries with identical string values have differing semantic meanings depending on the source column. For example, if there are multiple binary columns in the data table, we want to differentiate between `true`/`false` in each column. Numeric features are tokenized by quantizing the values into 32 bins (separated by quantile). The entries are not sub-word tokenized into word pieces as with common language tokenizers but instead are treated as atomic tokens similar to the word level tokenization algorithm (i.e. split by column). The result is that each processed event ($x_j$) with $k + 1$ features is tokenized into exactly $k + 1$ tokens such that

$$x_j = [f_0^{\text{meta}}, f_1^{\text{time}}, f_2, ..., f_k] \Rightarrow [t_0^{\text{meta}}, t_1^{\text{time}}, t_2, ..., t_k]$$

   where each $t_i$ is the tokenized version of $f_i$.

2. Events are grouped by the meta parameter and ordered by time into sequences of $l$ events

$$X_n = [x_1, ...x_l] = [f_{1,1}^{\text{time}}, f_{1,2}, ..., f_{1,k}, ...f_{2,1}^{\text{time}} ...f_{l,k}]$$

   For sequences with less than $l$ events, samples are right-padded with 0s so that each sequence $X_n$ is a single sample with exactly $k * l$ tokens that are ordered temporally and share a meta feature.

3. We use the same absolute positional encoding that was used for GPT, another decoder transformer (Radford et al., 2019). Therefore, since we do not designate the temporal feature time with special embedding but rather as a normal feature, there can be more than one temporal feature in the table, but there should be a time-ordered way of arranging the records.

We train STEP from scratch in the exact same fashion as GPT-style, decoder-only language models (Radford et al., 2019). We employ a standard next token prediction objective via cross-entropy loss and causal masking. Note that another benefit of having separate vocabularies for each column is that each token embedding naturally encodes its intra-record positional information since specific words only appear in specific locations. When the order of columns is the same for every row, the model predicts next tokens in the same order each time, however this restriction is not explicitly required by the model. As we will see, permuting the intra-record entries during training leads to other advantages.

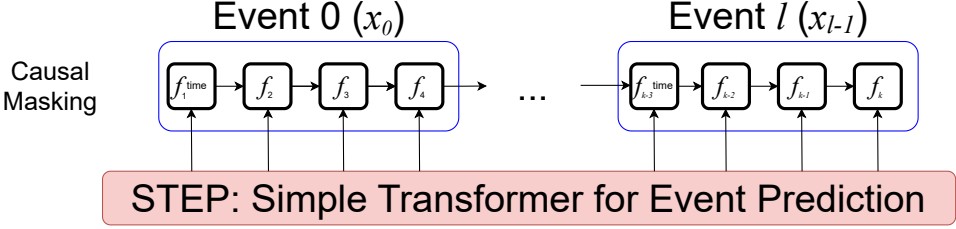

Figure 2: **STEP Event Processing.** STEP is a decoder-only transformer model, which means that each event is passed in sequentially with a causal mask applied.

### 3.3 DATA PREPROCESSING

A key step in our training pipeline is converting event data into text tokens to train our decoder-only transformer. We transform the columns of the table by respective data type: categorical values are kept as is, but numeric values are bucketed into 32 distinct buckets per column. Then we pretokenize the data by splitting splitting on column (rather than whitespace or digits as is common with traditional LLM pretokenization schemes). We train a word-level tokenizer on the data with a maximum vocabulary size of $60,000$ words. In contrast to Byte-Pair Encoding, word-level tokenization is a top down approach to building a vocabulary which simply takes the words in the corpus (in this case each individual cell is a "word") and uses the most common of these words as the vocabulary. While it would be possible to further split the cells into tokens for the model to learn information about the actual string values within the cell, we choose word-level tokenization as it ensure that each event (and therefore sequence of events) has the exact same number of tokens. Additionally, we tokenize the data such that each column in the table has a non-overlapping subset of the vocabulary to ensure that words that have the same values are perceived differently by the model (i.e. `True` has a different meaning in column `IsFraud` vs. column `IsOnline`). Each sequence is composed of $l$ consecutive records by the same user where $l$ is a dataset specific hyperparameter. Each record is separated by a token that delineates new events within the sequence.

In order to capture the temporal nature of each event, we replace the absolute timestamps with "time since last" to capture the relative time between each record. This column is treated as a normal feature (unlike Zhang et al. (2023) that uses this information as a special positional encoding).

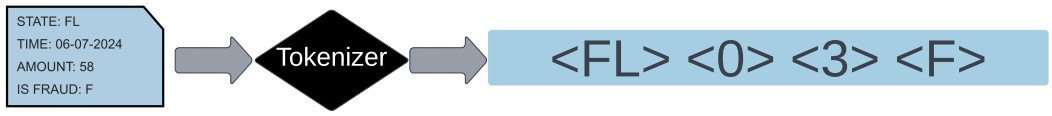

Figure 3: **Event Data Preprocessing pipeline**

### 4 EXPERIMENTAL SETUP

The data preprocessing required for STEP is relatively simple compared to traditional tabular processing pipelines – we employ an out-of-the-box word-level tokenizer and a standard causal language

model training setup. STEP uses the same architecture in each setting except for the context length, which is a function of the number of events per sequence and the number of features per event. Additionally, because we use a word-level tokenizer the maximum vocab size is 60000, but the actual vocab size can be much less and that directly affects the size of the embedding layer. For all datasets, STEP is trained using a next-token-prediction objective but is not finetuned on some downstream task. Rather, STEP has the flexibility to be used to for downstream tasks without the need for finetuning depending on which information is masked when passed to the model. Despite its lack of bells and whistles, we demonstrate the performance and flexibility of STEP compares favorably to prior work. See table 2 in A.4 for a full list of hyperparameters for our experiments.

## 4.1 DATASETS

We consider the 3 datasets that appear in prior work as well as 2 additional datasets and examine several tasks on each dataset. For more information see Appendix A.4.

### 4.1.1 SYNTHETIC CREDIT CARD TRANSACTIONS

The Synthetic Credit Card Transaction dataset was created by Padhi et al. (2021) to demonstrate the capabilities of TabGPT. It is comprised of credit card transactions ordered by user, card, and transaction time and contains information about the transaction including transaction amount, merchant city, merchant name, MCC, and a label indicating if the transaction is fraudulent or not. Prior works (Padhi et al., 2021; Zhang et al., 2023; Luetto et al., 2023) consider a task where the model predicts if the last transaction in a sequence is fraudulent or not based on prior transactions. In this work, we use sequences of length 10, where each sequence only contains transactions for a single user. Differing from prior work, sequences do not contain any overlapping records (i.e. `stride = sequence length`) since we believe this can bias the results. Additionally, we do not up-sample the positive labels to balance the classes.

The dataset contains approximately 24M transactions across 2000 users. We randomly separate 2% of the users into an evaluation set and train on the other 98% of users. Following prior work, we predict the final `IsFraud` label in the sequence; however, we also show that our model is flexible enough to handle other tasks and setups (see Section 4.3).

### 4.1.2 AMAZON PRODUCT REVIEWS

The Amazon Product Review dataset is a collection of reviews separated by product type (Ni et al., 2019). We analyze the "Movies and TV" and "Electronics" data subsets for comparison to prior work. We consider the `5core` subset of these categories, which only contains reviews by users who have submitted at least 5 reviews. The "Movies and TV" and "Electronics" datasets contain 3.4M and 6.7M reviews respectively, with columns indicating user, product, review score, time, and whether or not the user is verified or not. In these datasets, users rate products 1 to 5, however prior work converts this into a binary classification problem with ratings of 1, 2, or 3 falling into class 0 (unfavorable) and ratings of 4 or 5 constituting class 1 (favorable). As with the synthetic credit card transactions dataset, we consider a sequence length of 10 and predict the last review in each sequence.

### 4.1.3 CZECH BANK LOAN

The Czech Bank Loan dataset comes from PKDD'99 (Berka, 1999) and is comprised of real bank data for $\sim$4,500 clients across approximately 1M rows. It contains fields relating to client bank account information as well as loan information and can be used to predict if a client will default on a given loan. As opposed to the previous datasets, we use a much longer sequence length of 100 events for this task and a larger learning rate of $5 * 10^{-5}$. We believe we see better performance on longer sequences because the data comes from multiple data sources (loans and transaction events), so more events are needed to learn a relationship between the two. The primary goal of this task is to predict if a client will default on a given loan by looking at the events that occur over the life of that loan.

### 4.1.4 CLIENT CHURN

The Client Churn dataset (dllllb, 2024) is financial dataset that is used for the task of predicting if a client will leave the bank. The data consists of client transactions that are time-ordered and grouped

by user. The model is trained on the binary classification problem of predicting client outflow during the sequence. This dataset consists of approximately $10,000$ users and about $500,000$ total rows.

## 4.2 BASELINE COMPARISONS

We consider the work of TabFormer (Padhi et al., 2021), UniTTab (Luetto et al., 2023), and FATA-Trans (Zhang et al., 2023). Padhi et al. (2021) present TabFormer, which introduces the concept of using field transformers to learn row embeddings, which can be used in finetuning a model to predict the final label in a sequence. Additionally, Padhi et al. (2021) produces the widely used synthetic credit card transaction dataset by using a rule-based generator.

Luetto et al. (2023) introduce UniTTab, includes an additional linear layer between the field transformer and the sequence transformer which handles datasets with multiple types of events. Additionally, they employ a combinatorial timestamp and label smoothing to improve performance.

FATA-Trans (Zhang et al., 2023) uses a similar field transformer to encode each row, but includes time-aware positional embeddings, and field-aware positional embeddings to distinguish between static and dynamic features. They do extensive pretraining and demonstrate multiple ablations.

Each of these methods introduces performance improvements at the expense of increased complexity and rigidity. These sophisticated embeddings and training techniques marginally improve performance on this class of data but lack the flexibility to adapt to new prediction or modeling tasks on the fly. On the other hand, decoder-only foundation models are known to be adaptable to a diverse set of tabular data tasks (Gruver et al., 2023) in a zero shot setting despite tabular data containing a unique structure that differs from natural language. However, despite their flexibility, autogregressive LLMs underperform task specific models, but their ability to adapt to this setting serves as motivation that applying causal models to event data is a natural area for exploration. We will see that this simple approach can actually outperform highly engineered existing methods as well as current state-of-the art foundation models in a zero shot setting.

Additionally, we compare STEP against `Llama-3-8B-Instruct` (AI@Meta, 2024), which we access via the Huggingface API. We convert the table into text of the form `<header1>:<value1> <header2>:<value2>...[ROW]` and prompt the model for the last label in each sequence, restricting its output to only valid tokens. Results are included in table 1 (see A.2 for more details).

## 4.3 EXPERIMENT VARIATIONS

We now define our experimental setup. For each experiment, we train models with 5 different random seeds. For all datasets, we reserve 2% of each dataset as the test set, leaving 98% of the data to be used for training. We do this for each training run, meaning that the results are averaged over the 5 different evaluation splits. During training, we optionally mask some information about labels related to our desired task. For the credit card, Czech loan, and churn datasets, we mask the labels for all events as we likely do not know the labels of previous transactions ahead of time. For example, we do not know if previous transactions in the sequence are fraudulent when the model predicts if the final transaction is fraudulent. For amazon reviews, scores are likely immediately accessible after giving a review so we do not mask previous scores in the user sequence.

### 4.3.1 PREDICTING THE LAST LABEL IN A SEQUENCE

First, we show that in the setting considered by prior work (predicting the last label in the sequence), STEP outperforms the benchmarks. Events are considered in non-overlapping sequences. In this setting, the model is tasked with predicting the label for the final event in each sequence.

### 4.3.2 RANDOMIZING FEATURES WITHIN EACH EVENT

Because the features of an event all "occur" at the same time, we want to remove the causal bias inherent in decoder-only models. That is, for a given event, we want to break the intra-row dependency of the features from column 1 preceding column 2, column 2 preceding column 3, etc. The affect of this causal bias has been explored in the literature (Uria et al., 2014; Germain et al., 2015). In order to break this dependency, we train step with the ordering of features within a given event randomized so that the model learns not to expect a specific ordering of columns.

Prior work has primarily utilized MLM objectives, which are able to mask features *within* the input rather than just at the end. We achieve similar behavior by randomizing the order of the features within an event during training, which allows the model to predict any subset of a row's features using any of other features in addition to any number of prior transactions in the sequence.

### 4.3.3 HANDLING MISSING OR PARTIAL INFORMATION

A natural consequence of the model's ability to handle events with the features occurring in any order is that there is no set "label" during training – at evaluation time any feature can be predicted. Further, predictions can be made with any subset of features by simply passing in the known features for an event and predicting the missing ones. Similarly, while the model is trained with a fixed number of events in its context, it can make predictions with fewer events seen. We explore the trade-off between model performance and the amount of information seen by STEP.

## 5 RESULTS

In this section, we show STEP's performance across experiments and datasets. By training with these augmentations, STEP is able to handle situations where different subsets of information are received when making predictions at inference time. All results are shown by averaging over 5 random seeds.

### 5.1 STEP SURPASSES EXISTING BASELINES AT LAST LABEL PREDICTION

We first consider the setting described by prior work: we predict the last label in a sequence of records. Specifically, we show that when trained without randomization, STEP outperforms baselines without the need for finetuning. See table 1 for results.

| Dataset | STEP (w/o Random)[1] | STEP [2] | TabBERT[3] | FATA-Trans[4] | Llama[5] |
|---|---|---|---|---|---|
| Credit Card | $0.998 \pm 0.002$ | $0.981 \pm 0.009$ | 0.999 | 0.999 | 0.621 |
| ELECTRONICS | $0.771 \pm 0.007$ | $0.767 \pm 0.004$ | 0.710 | 0.721 | 0.693 |
| MOVIES | $0.853 \pm 0.006$ | $0.852 \pm 0.006$ | 0.796 | 0.806 | 0.753 |
| CZECH LOAN | $0.942 \pm 0.031$ | $0.935 \pm 0.045$ | 0.857 | | 0.499 |
| CHURN | $0.763 \pm 0.011$ | $0.752 \pm 0.005$ | | | 0.502 |

Table 1: **AUC scores for various models on label prediction task.** STEP is our full implementation with column randomization during training. STEP w/o random is a comparison to prior methods that train without randomization. Error values on AUC scores are calculated over 5 training runs.

### 5.2 TRAINING WITH RANDOMIZED FEATURES UNLOCKS NEW CAPABILITIES AT LITTLE COST

Next, we take advantage of STEP's flexibility to explore how randomizing the order of the features within an event during training allows the model to handle missing or partial event data at evaluation time. We show that by training in this way, STEP can receive any subset of the input sequence and make predictions on any values in the current or future records. While randomization marginally hurts performance, this ablation gives STEP the ability to handle missing and partial information (as shown in section 5.3). The second column of table 1 shows STEP performance with randomization.

### 5.3 STEP CAN HANDLE MISSING AND PARTIAL INFORMATION

Figure 4 shows STEP's performance for various sequence lengths and position labels. When the label is in position 0, and there is only one event in the sequence, the AUC score is .5 representing a

---

[1]See 4.3.1
[2]See 4.3.2
[3]Reported in Padhi et al. (2021)
[4]Reported in Zhang et al. (2023)
[5]See Appendix A.2 for more details

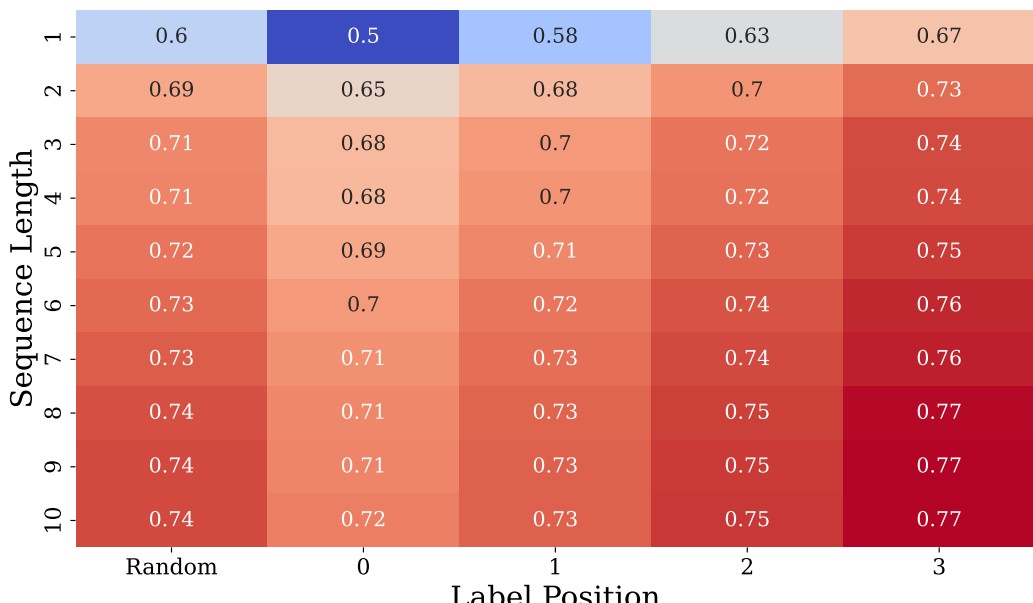

Figure 4: **Sequence/Label position trade off.** AUC scores for STEP across various sequence lengths and label position locations for the Amazon Electronics Dataset.

random guess. As the sequence length grows and the label is moved later in the sequence STEP's performance improves. For example, if the model is predicting the tenth event and the label is at the end of the record (i.e. positions 0,1,2 have already been seen by the model) the model has the best performance. This confirms what is intuitively true: the more information STEP has, the better it performs, but it is flexibly able to handle missing or random information without additional finetuning. See appendix A.1 for additional results.

## 6    DISCUSSION AND OUTSTANDING CHALLENGES FOR SEQUENTIAL TABULAR DATA

The baseline we propose in this work is flexible and can accommodate a variety of use-cases, but our work leaves several open problems:

**Multiple event types.** The datasets we use have the property that all events possess the same features. In many real-world applications, events may be of various types, each with their own features. For instance, a single user's history may include credit card payments, bill payments, and ATM transactions. Building models that can ingest and make predictions on data with multiple event types may require appropriate architectures or positional embeddings as well as training procedures. One benefit of our decoder only method is that it should be able to easily handle heterogeneous event data by simply adding special tokens indicating data source.

**Long context lengths.** In our experiments, we limit the number of tokens in a model's context by limiting the number of events. Other tasks may require loading many more events or features into the context. A disadvantage of the transformer architecture we employ is that the cost of attention increases quadratically in the context length. Future works may consider modifying the architecture to accelerate inference over long context lengths, for example straight-forwardly adopting existing tools from the language modeling literature or engineering new tools to select a subset of features per event or to select a subset of historic events that are useful for making predictions.

**Formatting model inputs.** One way to reduce the number of tokens in the context is by converting each event into a single token embedding, for example using a small embedding network, thus dividing the context length by the number of features per event. However, such techniques may

entail either different formats for input and output or may require predicting all attributes of an event simultaneously. If we naively predict an entire event simultaneously, we may not be able to compute joint probabilities of event attributes, which are required for some applications, such as modeling future events. For example, one approach might involve first embedding each event into a single token embedding using an embedding network, predicting the embedding corresponding to the next event, and then decoding that embedding autoregressively into the associated column entries. This approach would simultaneously reduce the context length while still enabling us to compute joint probabilities, but on the other hand would be more complexity with additional moving parts.

**Diverse benchmark datasets.** We focus on several popular datasets in our experiments, but applications of sequential tabular data are wide ranging, spanning scientific applications, finance, retail, recommendation, and more. An impactful area for future work is to collect diverse datasets and compile them into a systematic benchmark to improve the rigor of method comparisons. A future benchmark might also include datasets with multiple event types as described above.

**Limitations.** A major limitation of our experimental setup is the small number of publicly available datasets in this field, compared to the broader tabular data literature where vast numbers of datasets are in popular use. We endeavored to add multiple datasets and baselines but currently a primary challenge is the lack of publicly available datasets and easy to use baselines. We hope to expand and diversify datasets in future work and caution practitioners to carefully benchmark methods on their own data. We also only support categorical and numerical features, but in principle there is no reason to only consider these kinds of features. Real-world events might have text or graph descriptors as well.

Additionally, while our implementations are relatively straightforward, we view the simplicity of our methods as a feature of STEP. Prior work attempted to implement architectural and data ablations but we believe that demonstrating that simplicity is sufficient for superior results is a contribution of our method.

**Ethics.** Sequential tabular event data is common in financial settings where user information can be particularly sensitive. While this should be a consideration for anyone deploying a model like STEP, for this paper we used only publicly available datasets on synthetic or anonymized users.

**Reproducibility.** We have taken steps to make this work as reproducable as possible. All datasets described in section 4.1 are publicly available and we have provided cited each data source. A large part of our modeling pipeline is the event preprocessing that is described in section 3.3. Lastly, the code used to train STEP is publicly available on our github[6].

Instructions for how to run the code are included in the README.

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

# A  APPENDIX / SUPPLEMENTAL MATERIAL

## A.1  ADDITIONAL RESULTS

### A.1.1  PARTIAL MASKING

Rather than masking all of the labels of the previous events, we show the performance of STEP with partial masking. In this scenario, all labels are masked with 25% probability and 50% are masked with 25% probability. Training step in the fashion makes it robust to partial masking at evaluation time. See figure A.1.1.

### A.1.2  LABEL POSITION, SEQUENCE LENGTHS

Included are the heatmaps for performance across varying label positions and sequences lengths for other datasets.

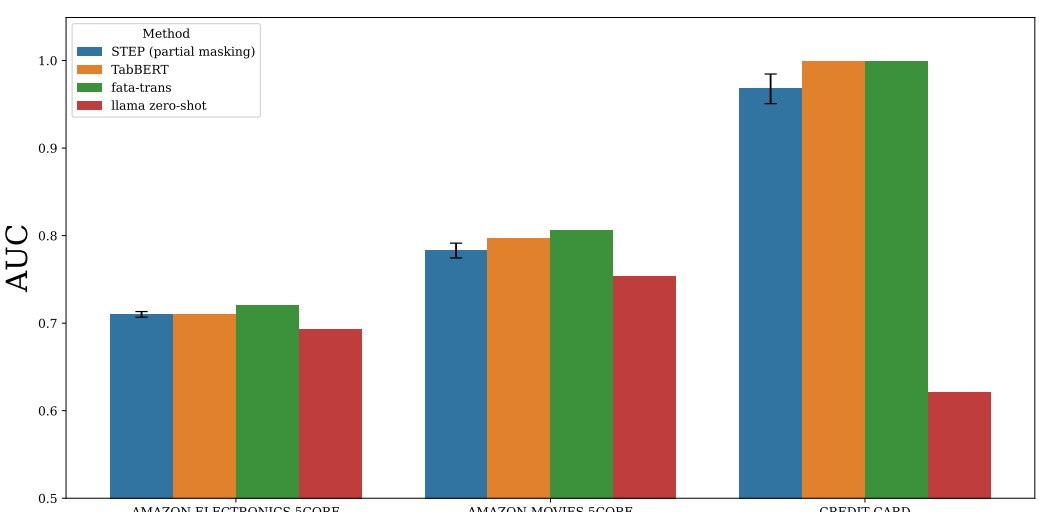

Figure 5: **STEP with partial masking**

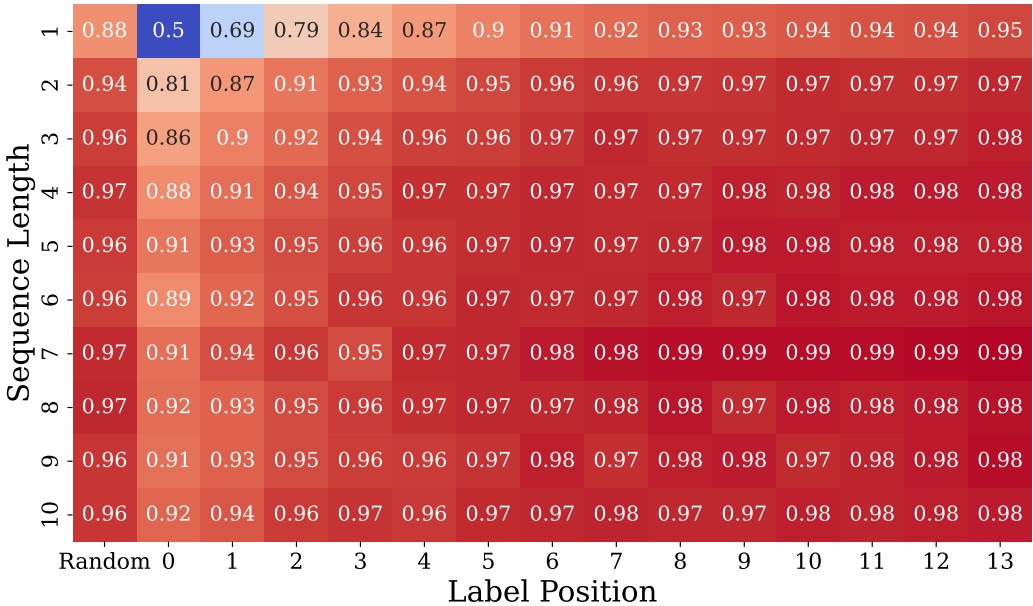

Figure 6: **Sequence/Label position trade off.** STEP performance across various sequence lengths and label position locations for the Synthetic Credit Card Dataset.

## A.2 PROMPTING LLAMA 3

To compare STEP's performance against a state-of-the-art LLM we prompt `Llama-3-8B-Instruct` (AI@Meta, 2024) and evaluate its zero-shot performance across the datasets and tasks. In order to prompt Llama, we first need to convert the data into a text form that the model can understand. This is done using the following system prompt:

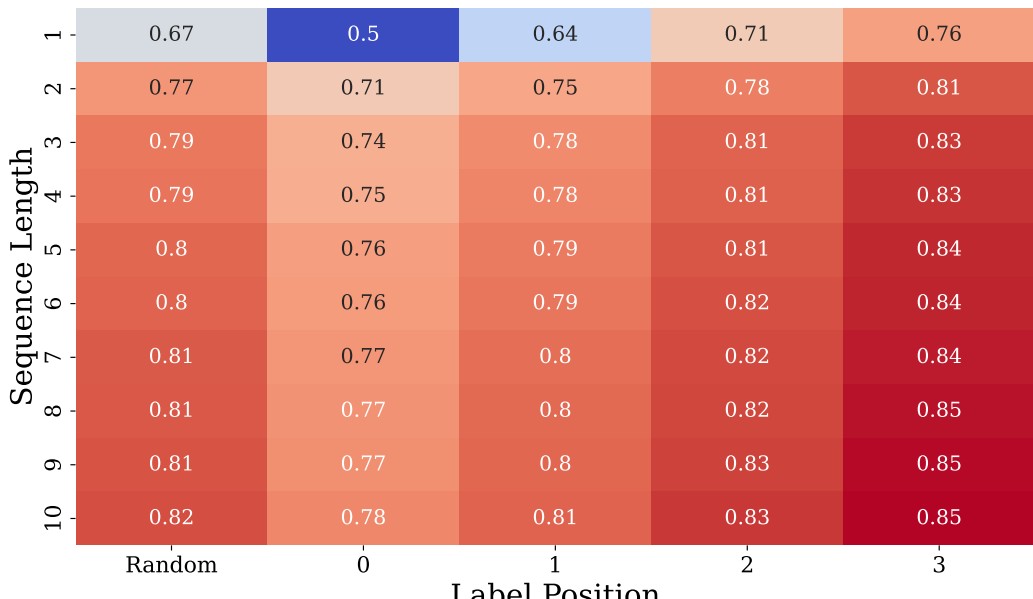

Figure 7: **Sequence/Label position trade off.** STEP performance across various sequence lengths and label position locations for the Amazon Movies Dataset.

**Figure 8: Llama prompting**

```
You are a helpful AI assistant that is good at analyzing
sequential tabular data.  The user will give you tabular data
in the form '<column name>:<value>' with '[ROW]' indicating
the end of a row.  You will be given the next column name and
are tasked with predicting the corresponding value.  Please
only respond with the next value.
```

An example event sequence for the churn dataset is:
```
trx_category:POS MCC:5331 channel_type:type1 currency:810
amount:84000.0 total_minutes_from_last:0 [ROW]
trx_category:DEPOSIT MCC:6011 channel_type:type1 currency:810
amount:74500.0 total_minutes_from_last:662 [ROW]
trx_category:DEPOSIT MCC:6011 channel_type:type1 currency:810
amount:10000.0 total_minutes_from_last:1 [ROW]
trx_category:POS MCC:5331 channel_type:type1 currency:810
amount:78000.0 total_minutes_from_last:10857 [ROW]
trx_category:DEPOSIT MCC:6011 channel_type:type1 currency:810
amount:78000.0 total_minutes_from_last:808 [ROW]
trx_category:POS MCC:5411 channel_type:type1 currency:810
amount:428.0 total_minutes_from_last:13592 [ROW]
trx_category:POS MCC:5331 channel_type:type1 currency:810
amount:10000.0 total_minutes_from_last:2880 [ROW]
trx_category:DEPOSIT MCC:6011 channel_type:type1 currency:810
amount:10000.0 total_minutes_from_last:867 [ROW]
trx_category:POS MCC:763 channel_type:type1 currency:810
amount:168.0 total_minutes_from_last:52413 [ROW]
trx_category:POS MCC:5331 channel_type:type1 currency:810
amount:84000.0 total_minutes_from_last:5760
User Churn During Period:
```

In the sequence there are 10 events. The model is then asked to label the sequence as `True`/`False` depending on if it predicts client churn or not:

Importantly, the output of the model is restricted to only the tokens representing `True`/`False` for fair comparison to STEP which is prompted specifically to answer the churn question.

## A.3 COMPUTE USAGE

The code for this project is run in 4 states:

1. Train tokenizer
2. Tokenize data
3. Train the model
4. Evaluate the model

Training the tokenizer and tokenization are both cpu intensive tasks that do not require much use of GPUs. As such, these were run on CPU only computing clusers. Training and evaluating the model necessitated the use of GPUs, for which we use Nvidia a100's with varying memory depending on the side of the dataset. Training STEP on the synthetic credit card transaction dataset and the Amazon review datasets took 24 and 12 hours for 20 epochs respectively. The training for the Churn and Czech Loan datasets took under 1 hour each.

## A.4 HYPERPARAMETERS

We include final hyperparameters in table 2. Additionally, our code release includes each separate experiments and corresponding hyperparameters. We came to these hyperparameters through trial and error across multiple runs, however we found that for many of these the results were not particularly sensitive to our choices.

| Dataset | Credit Card | Electronics | Movies | Czech Loan | Churn |
|---|---|---|---|---|---|
| **Epochs** | 20 | 20 | 20 | 50 | 50 |
| **Seq Length** | 10 | 10 | 10 | 100 | 10 |
| **LR** | 1.00E-05 | 1.00E-05 | 1.00E-05 | 5.00E-05 | 1.00E-05 |
| **Batch Size** | 64 | 64 | 64 | 16 | 16 |
| **Mask Labels** | TRUE | FALSE | FALSE | TRUE | TRUE |
| **Test Split** | 2% | 2% | 2% | 2% | 2% |
| **Events** | ~24M | ~6.7M | ~3.4M | ~200,000 | ~500,000 |
| **Sequences** | 2.5M | ~700,000 | ~350,000 | ~2,000 | ~50,000 |
| **Users** | 2,000 | ~70,000 | ~30,000 | ~800 | 5,000 |
| **Features** | 13 | 3 | 3 | 7 | 6 |
| **Model Params** | 178M | 178M | 178M | 86M | 86.6M |
| **Vocab Size** | 60,000 | 60,000 | 60,000 | 100 | 468 |
| **Embedding Size** | 768 | 768 | 768 | 768 | 768 |
| **Hidden Size** | 1024 | 1024 | 1024 | 1024 | 1024 |
| **Attention Heads** | 8 | 8 | 8 | 8 | 8 |

Table 2: Summary of Experimental Configurations for Different Datasets

