# OpenReview forum: "A Simple Baseline for Predicting Future Events with Auto-Regressive Tabular Transformers"
_ICLR.cc/2025/Conference — Submitted to ICLR 2025_

### Official Review · Reviewer_XeVZ · 2024-10-23

**Soundness:** 2
**Presentation:** 2
**Contribution:** 1
**Rating:** 3
**Confidence:** 4

**Summary:**

This paper proposes to model event streams using a standard autoregressive, decoder-only Transformer (like the ones used in modern LLMs).  Event streams are unique compared to time series data in that the events may occur at arbitrary timestamps.  It is also notable that in event stream applications, key covariates/features may be missing within the context window.  In this paper, the approach is to “flatten” multivariate tabular event data into a single sequence of tokens, where features/fields of individual events (individual “rows”) occur as a sub-sequence (of tokens) within a larger, ordered-by-time sequence-of-subsequences over all the events.  In order to model the arbitrary arrival times of events, this paper proposes encoding the inter-arrival duration as a numeric feature (binned into one of 32 bins for tokenization).  To handle specific missing features, this paper proposes to mask all of the target features in the context window during training (or the system could be re-trained from scratch with partially-masked labels, although it does not perform as well on the evaluation tasks if this is done).  Finally, to enable predicting arbitrary features/fields of individual events using a single model, the paper proposes creating multiple versions of training sequences, each with the fields in a random order WITHIN each event sub-sequence.  A single model can be trained over all the randomized orderings collectively.  The proposed method shows competitive accuracy with two prior approaches on a few prior tasks.

**Strengths:**

Using vanilla Transformers to model event data is a good idea.  LLMs are already being trained on multi-modal data, either implicitly by consuming data of different types (including tabular data) from the web, or explicitly by being fed speech, image, video, and other modalities during training.  Prior work has shown LLMs can already be used to predict time series data out-of-the-box without any special fine-tuning.  It seems modeling events will be within the capabilities of foundational models in short order.  As such, this paper is on the right track.

The writing within each section is fairly clear.  Figure 1 is helpful.  Figure 4, 6, and 7 are good things to test.

**Weaknesses:**

Overall, it seems like if this approach is being positioned as a simple baseline that nevertheless works quite well on a range of tasks compared to other approaches, then the empirical evaluation needs to be much more extensive and rigorous and convincing.  I also feel like it’s a missed opportunity: since STEP uses its own vocab and whatnot, it doesn’t seem capable of leveraging *pre-trained* LLMs (e.g., fine-tuning them to work on event stream data).  It would have been useful to investigate the possibility of training event predictors using tokenization, etc., that is COMPATIBLE with LLMs, and maybe even use the LLMs directly to do the prediction, along the lines of what Gruver et al. did for time series forecasting.

Experimentally, it was difficult to put the results in context or understand what was done:

- What is the random baseline on each task?  I assume Table 1 is accuracy?  Wouldn’t precision/recall make more sense for a low-frequency detection task like detecting fraudulent transactions?  Especially since no steps were taken to address data imbalance. The results suggest STEP is much, much worse than TabBERT or FATA-Trans on the credit card data (19x the error), but the way the results are presented do not make this very clear, nor suggest what is going on (does STEP predict “non-fraudulent” in all cases?  We don’t know from Table 1).
- How are the error bars determined in Table 1?
- It seems like STEP is very sensitive to the things that give it unique capabilities.  E.g., when randomizing, the error rate on the credit card data increases significantly.  Also, Section A.1.1 shows that STEP with partial masking is worse than the baselines across all tasks.  This sensitivity isn’t really explained in the paper.  The brittleness of prior work is emphasized in the paper abstract, but STEP also seems hard to recommend for use out-of-the-box.

Also, as this is event stream prediction, it’s notable there is no evaluation of predicting the inter-arrival times of events:

- How accurately can STEP predict time-to-event?  How important is time as a feature?  What if you removed it entirely?
- What do you lose by not knowing the absolute time? (e.g., not knowing whether an event occurs on a weekend or not)
- Is 32-bins enough for numeric data?  What about 64?  Or 16?

Some other natural questions are not considered or not explained clearly:

- “We came to these hyperparemeters [sic] through trial and error across multiple runs” – but there’s no mention of a separate dev/validation set for hyperparameter tuning. Did you tune the HPs on the test data?
- You mention “Event Stream GPT (ESGPT), an open-source library designed to streamline the end-to-end process for building GPTs for continuous-time event sequences” – but can’t we just use ESGPT as the baseline for event stream prediction?  Why do we need STEP, another baseline?  Does STEP work better than ESGPT for some reason?
- Transformers are known to be very “data-hungry”.  How does accuracy here depend on the amount of training data?
- Why does this method improve over the prior art on ELECTRONICS and MOVIES?  Are we sure we used the exact same train/test splits?  Did they really both use exactly length-10 sequences, the same as in this work?  Do we use the same data splits?
- How could we handle predicting a label on an event based partly on future *events*?  E.g., say someone purchased jewelry with the credit card (maybe fraud?) and then next they purchase a coffee, then got a subway token, etc.  Could those future purchases be used to influence our assessment of the jewelry purchase?

The paper is also a bit disorganized.  For example, the baseline systems are described in the methodology section, but not in experimental details, except the Llama baseline, which is introduced in “Results”.  Data preprocessing is mentioned in 3.3 but also the same example is used in 4.2.  So I found myself having to jump around a bit to get the details.  It took me a long time to go back and find where the paper mentions that it uses “time since last event” rather than the absolute timestamp (it wasn’t in 3.1 and 3.2, which discuss encoding time, but rather it was in 4.2).

Finally, I find this work lacks an understanding of what an autoregressive generative model is, i.e., a joint model over the data, factorized in a certain manner.  Autoregressive transformers are just used here as a tool.  So it’s a very applied solution.  A more theoretical understanding could help connect things here to prior generative models.  E.g., the idea of being able to impute missing data in different orders has been explored before – e.g., it’s similar to what is done in MADE/NADE – See Uria et al., https://arxiv.org/abs/1310.1757 for the re-orderings, also used in MADE, Germain et al., https://proceedings.mlr.press/v37/germain15.html.

Things that do not affect the review scores:

- Normally figures in papers are so small, but here it seems like Figure 4 is *unnecessarily large*
- Regarding the synthetic credit card data, in the TabGPT paper it says, “the transactions are created using a rule-based generator where values are produced by stochastic sampling techniques, similar to a method followed by [19].”  But in this paper, it suggests this dataset was made via TabGPT itself: “additionally, Padhi et al. (2021) produces the widely used synthetic credit card transaction dataset by training a causal decoder on top of TabBERT (appropriately called TabGPT).”
Typos start creeping up near the end of the paper:
- “task of predicting if a client with with leave the bank”
- “The each record is separated by a token that delineates new events within the sequence.”
- “we explore how masking some or all of the prior event label during allows the model”

**Questions:**

Note quite a few questions are noted above under Weaknesses.

“Additionally, because we use a word-level tokenizer the maximum vocab size is 60000” – please explain.

5.3 – if you pass in fewer events (e.g., shorter sequence lengths in Figure 4), I don’t get why the model would predict a label at all.  Like, didn’t we mask out all the labels for the first 9 events?  Wouldn’t it only predict a label for the 10th event?

Since you can randomize the order of the observed features, can you do better by averaging over multiple re-orderings or feature subsets?  E.g., say you are predicting column D and you’ve seen only A, B, C.  Can you predict A, C, B, <>, and B, A, C, <>, and C, B, A, <>, etc?  This also connects to the MADE/NADE work.

---

> ### Author Response · Authors · 2024-11-22
> **Response to Reviewer XeVZ**
>
> We thank you for your thorough review and address your questions below.
>
> The primary reason for this being described as a "simple baseline" is due to its flexibility and lack of over engineering compared to prior work on event data.
> The flexibility of our method is described in 4.3: We show that the label column (i.e. the column that we are predicting) can be in any location in the table and similarly, we can change which column serves as the label (i.e. predicting time_to_event or is_fraud) at runtime without the need for any column specific finetuning
> While we agree that it would be possible to finetune an LLM to perform these tasks, in this work we are primarily demonstrating that you could use a lightweight, autoregressive model to do the same thing. In table 1 we show zero-shot performance using an LLM (Llama3-8B) to show that it is not sufficient to use an out-of-the-box LLM for event prediction tasks.
> Ultimately our primary contribution is this:  There have been a limited number of attempts to use transformers for event prediction tasks, but all of them have used convoluted architectures, features, training regimes, etc.  Additionally, all of them have used encoders to encode events before using a second model (often a decoder) to predict future events.  In our work, we show that by using a simple decoder-only transformer and removing the special event prediction related features we are able to outperform current methods.
> Our work serves as a baseline in its flexibility across a number of prediction tasks and its simplicity by removing intricate architectural features.
>
> In response to your questions about metrics and Table 1: Table 1 shows AUC scores and the errors were calculated by averaging over 5 random seeds.
> We have added information about both of these things to the caption for Table 1.
> Randomizing does decrease performance slightly, but does not dramatically increase the error rate (in some cases the error is actually smaller when randomizing).
>
> In terms of decreased performance when masking labels, while it is true that STEP with masking underperforms baselines, it would be better to compare STEP with masking to baselines with masking.
>
> Additionally, given that time-to-event is just another feature it can be predicted in the exact same way as the label feature. Similarly, it can be removed from the table at run time and STEP can make predictions without it (this is true with time-to-event as well as with any other feature).
> Additionally, we agree that there should be further research into the correct way to handle numeric values.
> The idea to quantize the numeric features was based on the decisions of prior work to do the same.
> Prior work primarily used 10 buckets, we decided to use 32 (although we tried 64 and saw similar performance).
> Ideally, numeric values would be embedded directly rather than bucketed but for the purpose of defining a baseline we felt that 32 buckets was sufficient given our comparison to prior work.
> We use exactly length 10 sequences as in prior work but we do not use the same train/test split (as those splits are not available). Additionally, in order to remove potentially getting a favorable train test split we use a method similar to 5-fold validation to calculate AUC scores.
>
> Regarding your questions about hyperparmeter tuning, we did not hold out a portion of the dataset for validation but rather we choose hyperparameters by running a series of tests on different portions of the data and averaging performance.  Specifically, we ran k-fold validation (with $k=5$) to determine the best hyperparameters before actually running all of our training runs.
> Similarly, we did not have a designated train/test split that we used for every run, but rather each run took a a unique split of the users into train/test sets to be trained on and then evaluated for that model.  The performance was then averaged across 5 seeds to avoid any anomalies that might occur from any specific split of the users.
>
> Regarding ESGPT as a potential baseline, while ESGPT is a valuable library, their work primarily focuses on data cleaning and flexibility of the MIMIC-IV dataset rather than focusing on a new method for handling this type of data. Additionally, while they provide some code to train a transformer on this specific domain they specifically consider encoder/decoder models rather than the decoder only models we consider. Further, they use intricate nested attention layers rather than our vanilla causal attention. Our work is differentiated in its simplicity.

---

> > ### Author Response · Authors · 2024-11-22
> > **Response to Reviewer XeVZ (cont.)**
> >
> > In response to your question about predicting labels based on future events as well as other potential ablations:
> > While this is not a use case we considered it would be possible for our framework to handle this. We would move the event in question to the end of the sequence. However this would result in a negative value for the "time since last" variable, which would have unknown effects on the performance of the model.
> > We are relying on the fact that in general, for event sequences, samples occur in a known ordering according to a temporal feature.
> >
> > The value of STEP is that it can handle missing information or fewer events in the sequence.  The more information it has the better it performs, be it by having more information about the event in question OR having more events in the sequence.
> > If the previous labels are masked the model still can make predictions based on other features from previous events.
> >
> > We have made the following changes in improve organization and presentation:
> > 1. moved the prior methods into a separate section
> > 2. moved the first paragraph of the Results section into the Experimental Details section (including the Llama baseline)
> > 3. Moved the Data preprocessing section (4.2) to the Methodology section (section 3)
> > 4. Added more details to the Experiment Variations section
> > 5. Added clarifications on how we produced the tokenizer:
> > In a word-level tokenizer, the size of the vocabulary is set a priori and each field is a token candidate. Then the most common tokens are selected (leaving some tokens as "unknown") until the vocabulary is full.
> > We have added this information to the experimental setup section.
> >
> > The goal of our work is to design and validate a simple and effective solution to modeling events using a decoder-only transformer.  We agree that theoretically unifying decoder-only transformer approaches with previous methods for NADE or even older statistical methods is an interesting opportunity for future work, but our focus is, as you say, on the applied problem of using a simple decoder-only transformer to model events.  We do agree that such works as the ones you brought up are relevant, and we have now updated our draft to mention them.
> >
> > Thank you for this clarification about TabGPT generating the synthetic dataset.  In the TabGPT paper it says "Specifically, we train a GPT model (referred to throughout as TabGPT) on user-level data from the credit card dataset in order to generate synthetic transactions that mimic a user’s purchasing behavior. This synthetic data can subsequently be used in downstream tasks without the precautions that would typically be necessary when handling private information."
> > And we took this to mean that the data provided was the result of this synthetic data augmentation.  We have adjusted the language in our paper to reflect your correction.
> >
> > Lastly, in response to your question about averaging over the ordering of the features: This is exactly what happens in the randomization, the event is fully randomized so each random seed that it is averaged over has a different ordering of prior features.

---

> > > ### Comment · Reviewer_XeVZ · 2024-11-23
> > >
> > > I thank the authors for their response to the review.  Your clarifications make sense.  Your discussion for handling sequences out-of-order is interesting to me, and also makes sense.  I think the proposed changes to the organization, presentation, and discussion of related work will make the paper stronger.  I still question whether you should tune your hyperparameters on the same data as that on which you evaluate (even if they are different random splits).
> > >
> > > I still feel the overall contribution, significance, and depth of this work is low by ICLR standards.  I would consider raising my review score slightly but note the granularity is very low in this range (there is no option for a rating of 4, for example).

---

### Official Review · Reviewer_gxkW · 2024-10-31

**Soundness:** 2
**Presentation:** 2
**Contribution:** 2
**Rating:** 3
**Confidence:** 3

**Summary:**

This paper proposes a simple yet flexible baseline model for tabular event data, utilizing standard autoregressive, LLM-style transformers capable of addressing a variety of use cases involving different data types and tasks.

**Strengths:**

The authors' intention to create a versatile baseline model tailored for tabular event data is commendable and has the potential to accelerate advancements in this field.

**Weaknesses:**

The experimental section is somewhat lacking. Beyond Table 1, the authors should conduct additional experiments to showcase various aspects of the proposed model and provide a more comprehensive comparison with existing models, which I believe is essential for baseline papers. Furthermore, Figure 4 conveys limited information, making it seem unnecessary.

**Questions:**

- In line 386, is the caption of Table 1 incorrectly labeled? Why is it titled 'Summary of experimental configurations for different datasets'?
- In line 327, there are two occurrences of 'with.'

---

> ### Author Response · Authors · 2024-11-22
> **Response to reviewer gxkW**
>
> Thank you for taking the time to review our paper.  We have responded to you points below:
>
> In response to your questions about our experimental details:
> Figure 4 shows one of the important variations of our method, namely that the model can handle the label in different positions and with different sequence lengths without additional finetuning. This type of flexibility is the primary contribution of our work.
>
> Would you be willing to elaborate more on what you think the experimental section is lacking?  In our work we have outlined that there are limited baselines for event prediction tasks and we aim to provide a framework that is more simple than the few existing baselines, but we do compare our method to theirs.
>
> Additionally, we have updated the title of figure 1 for additional clarity.

---

### Official Review · Reviewer_Lm89 · 2024-11-05

**Soundness:** 3
**Presentation:** 2
**Contribution:** 3
**Rating:** 5
**Confidence:** 3

**Summary:**

This work proposes a method, called STEP, for training decoder-only, LLM-style models for tabular data understanding tasks. The authors design a column-wise tokenization mechanism and use time-based data packing to preprocess rows of events in a table. They train the decoder with event features in arbitrary orders to simulate the masked language model (MLM) training used in other methods. Experiments across multiple datasets show that the proposed approach achieves good performance.

**Strengths:**

1. Design of using a decoder-only LLM with a causal language modeling objective to encode and predict tabular data is new.
2. The authors conduct experiments on a variety of datasets and tasks, where the method achieves better performance than two tabular transformer baselines.

**Weaknesses:**

1. Unclear Motivation: The motivation for using a decoder-only architecture for encoding-style tasks is not well-explained. The authors simulate MLM training with features in arbitrary orders, which raises the question of why an encoder-based architecture and traditional MLM training weren’t used directly. If performance was the factor, then the results in Table 1, where LLaMa performs the worst, might indicate that a decoder-type transformer isn’t optimal for these encoding tasks. While STEP performs better than the baselines, baseline performance is quite low. Were the baseline results reported after equivalent fine-tuning on the same amount of training data?

2. Vocabulary Use: It is unclear how separate vocabularies are used during training. Why not simply use a single tokenizer and a unified vocabulary by assigning different IDs to identical values from different columns?

3. Experiment Details: Key experimental details are missing. For instance, the configuration for the base LLM is not specified — is the model a pre-trained LLM or randomly initialized? If pre-trained, an ablation study comparing performance before and after your training would have been necessary. Also, the vocabulary size of 60,000 tokens is ambiguous—does that mean STEP only has ~60,000 unique numbers?

4. Weak Baselines: There are more advanced models for handling tabular or structured data, such as GPT-4o, Claude, TableGPT, and code models like CodeLlama or CodeQwen. A more comprehensive comparison with these stronger baselines would be more convincing.

5. The proposed method requires separate training for each task or dataset, which is not general. Details on training time and GPU requirements are missing as well.

**Questions:**

Which decoder-only LLM is used for training? Is it a pre-trained LLM or a randomly initialized model? How many layers are in the model?

---

> ### Author Response · Authors · 2024-11-22
> **Response to reviewer Lm89**
>
> We thank you for your response and address your questions below.
>
> While prior work has primarily used encoder-type solutions, that does not mean this task requires the use of encoders.  In fact, we believe that our primary contribution is showing that a decoder-only model outperforms the encoder-based baselines.  Further, treating event prediction as a generative task gives the model the flexibility to not need to be finetunes on specific targets at runtime.
>
> In response to Llama underperforming in the zero shot setting indicating that decoder-only LLM's arent suited for event prediction: performance is the main factor we consider in our comparison, the Llama column in Table 1 is zero-shot performance and is used as a baseline demonstrating that our method outperforms foundation models.
> Of course, you could finetune an LLM for each task/dataset, but a primary advantage of STEP is the lightweight nature of the method which includes a smaller model, much less training, and no further finetuning for each task on the given dataset.
> Baseline results are reported from prior work and we agree that baselines results are limited.  We believe the lack of strong baselines is further proof that event prediction needs a more robust set of benchmarks and baselines.
>
> In response to questions about our separate vocabularies, we implement the separate vocabularies using a single tokenizer object. We specifically used the term "vocabularies" instead of "tokenizers" to show that the vocabularies for each column are non overlapping. It would be possible to use separate tokenizers per column but we chose to implement it as you said by having a single tokenizer, where different ID's can be assigned to the same string values.
> The difference between these two things is simply an implementation detail.  We have updated the paper to make this distinction more clear.
>
> Additionally, a vocab size of 60,000 was chosen by observing performance on a few different vocabulary sizes. We did not observe a material difference in performance when attempting other vocab sizes. It does not mean that there are only 60,000 unique numbers as the numeric values are quantized into 32 different buckets (this process is described in section 4.2 where we discuss data preprocessing). The choice to bucket numeric values rather than using the direct values themselves is in line with prior methods (although we agree that this would be a potential future area of research).
>
> In response to requesting additional clarify on experimental details: in section 3.3 we walk through the training setup. Our model is trained from scratch and is not a fine-tuned version of a pre-trained LLM. We have added additional information to this section to make it more clear.
>
> Regarding your question about additional LLM's as baselines: We are specifically discussing the event prediction modality, which is a subset of tabular/structured data and has been relatively underexplored. The primary difference between event prediction and general tabular domains is the addition of a temporal feature and a meta-feature (where events are grouped by the meta-feature). We discuss this distinction in section 1, but have added more information there for clarity.
> While we are not able to make a fair comparison to the closed-source models you mentioned (as we have no way of viewing the output logits to evaluate their performance), we do include a comparison to Llama3 as a baseline.
>
> Additionally, this method does not need to be fine-tuned for each task.  Because it is generative in nature it learns the joint distribution across the entire dataset and can predict any label at evaluation time.
> That being said, our primary goal was not to create a foundation model for event prediction as this would require much more publically available data (we discuss in section 6 that one of the things that makes event prediction difficult is the lack of publicly available training data like there is in time-series or general tabular domains).
> Additionally, we include compute and training requirements in Appendix A.3.
> Training and model hyperparameters are included in Appendix A.4.  Section 3.3 gives our methodology:
> ```
> We train STEP in the exact same fashion as GPT-style decoder-only language models (Radford et al., 2019). We employ a standard next token prediction objective via cross-entropy loss and causal masking.
> ```

---

### Meta-Review · Area_Chair_gR9v · 2024-12-10

**Metareview:**

The paper appears to be very straight forward for an ICLR submission though it may hide some nice ideas as pointed out in the discussion with the authors. Nevertheless, there is not enough in it for ICLR. There are also some design choices that need either a much better motivation or need to change in the next version (e.g., hyper parameter tuning).

**Additional Comments On Reviewer Discussion:**

There has been a bit of a discussion with mainly one reviewer which was very insightful. It would have been nice to hear from other reviewers as well though. However, none of the reviewers was excited about the paper in the first place and I think the authors understood from the reviews already that chances to get in will be low.

---

### Decision · Program_Chairs · 2025-01-22

Reject